# Breaking social media fads and uncovering the safety and efficacy of mouth taping in patients with mouth breathing, sleep disordered breathing, or obstructive sleep apnea: A systematic review

Jess Rhee[1], Alla Iansavitchene[2], Sonya Mannala[3], M. Elise Graham[1], Brian Rotenberg[1]*

1 Department of Otolaryngology – Head and Neck Surgery, London Health Sciences Centre, London, Ontario, Canada, 2 Department of Library Services, London Health Sciences Centre, London, Ontario, Canada, 3 University of Saskatchewan College of Medicine, Saskatoon, Saskatchewan, Canada

* brian.rotenberg@sjhc.london.on.ca

## Abstract

### Background

Social media has contributed to a potentially unsafe trend of nighttime mouth taping for individuals with mouth breathing, sleep disordered breathing, or sleep apnea as a home remedy to treat these issues. This systematic review is aimed to highlight any potential benefits or harms with this practice.

### Methods

A comprehensive librarian-designed literature search was performed using PRISMA guidelines. Using search terms, "mouth taping, adhesive mouthpiece, porous oral patch, surgical tape, breathing mouthpiece, sleep, microsleep, breath, breathing, or mouth breathing", MEDLINE, Embase, and Google Scholar were searched from February 1999 to February 2024. Covidence software was used for screening and data entry performed into a data collection sheet designed *a priori.*

### Results

Covidence software was utilized to screen 120 articles. After 34 duplicates were removed, 86 articles were screened by two independent reviewers. Sixty-two were excluded. Twenty-four went on to full text review and 10 met inclusion criteria with a total of 213 patients. Two studies showed statistically significant improvement in established markers of sleep apnea such as apnea-hypopnea index (AHI) or oxygen desaturations. Other studies showed that mouth taping offered no differences and even discussed potential risks including asphyxiation in the presence of nasal obstruction. Many studies excluded anyone with nasal obstruction or pathology.

**Data availability statement:** All relevant data are within the manuscript and its Supporting Information files.

**Funding:** The author(s) received no specific funding for this work.

**Competing interests:** The authors have declared that no competing interests exist.

## Conclusion

The social media trend of mouth taping for individuals with mouth breathing, sleep disordered breathing, or sleep apnea has been reviewed. Based on the data presented by these 10 different studies, it seems that there is a potentially serious risk of harm for individuals indiscriminately practicing this trend. Further studies are required to elucidate any clinical benefit this practice may have.

## Introduction

Obstructive sleep apnea (OSA) is characterized by interruptions of breathing during sleep and is considered the extreme end of sleep disordered breathing (SDB) [1]. OSA causes oxygen desaturation events and, depending on the severity, can cause long-term sequelae including hypertension and cardiovascular, pulmonary, and quality of life detriments [1,2]. Mouth breathing has been identified as a risk factor for OSA [1,2]. Mouth breathing also worsens OSA by narrowing the airway and increasing obstruction [1,2]. Sleep disordered breathing (SDB) or OSA in children is typically managed through adenotonsillectomy [3]. In adults, OSA is usually treated with a continuous positive airway pressure (CPAP) machine [2,4]. However, for many individuals, CPAP adherence is poor due to discomfort [4].

Mouth breathing occurs when either nasal or pharyngeal obstruction compels individuals to switch from nasal breathing to breathing through the mouth. Allergic rhinitis, adenoidal and tonsillar hypertrophy and deviation of the nasal septum are among the most common causes for mouth breathing [5]. Nasal obstruction and the resultant mouth breathing has also been implicated in SDB and OSA [6,7], with SDB considered to be both a cause and a consequence of nasal obstruction.

Many interventions have been considered to address mouth breathing. One such intervention that has increased in popularity likely due to social media trends is the phenomenon of mouth taping. This involves participants maintaining mouth closure by occlusion with tape while sleeping to prevent mouth breathing. Participants allege benefits ranging from better sleep quality to anti-aging properties to improvements in dry mouth, bad breath, and concentration [8], but lack of concrete evidence gives rise to concern about this practice both from a safety and effectiveness perspective.

The aim of this study was to investigate the literature to determine the effects of mouth taping on SDB and OSA to assess if this practice carries meaningful benefit and/or risk of harm.

## Methods

A comprehensive search strategy was devised with an assistance of a clinical librarian (AI) with experience in conducting searches in electronic databases. Adhering to Preferred Reporting Items for Systematic Reviews and Meta-Analyses (PRISMA) guidelines, the systematic search strategy was tailored to our predefined inclusion and exclusion criteria and conducted using MEDLINE® and Embase® (both via the OVID platform) electronic databases. The web-based search engine Google Scholar

was searched to identify additional potentially relevant non-indexed articles in bibliographic databases. References of all studies identified as applicable for inclusion were reviewed for additional articles relevant to our systematic review.

Systematic literature searches were carried out from February 1999 until February 2024. To identify relevant studies, we utilized a sensitive search strategy comprised of the following search terms (using combinations of subject headings (i.e., MeSH in MEDLINE) and keywords): mouth taping, adhesive mouthpiece, porous oral patch, surgical tape, breathing mouthpiece, sleep, microsleep, breath, breathing, or mouth breathing with further filtering to adverse effects. English language restriction was applied. The search strategies were modified using appropriate thesaurus terms and fields suitable for each database.

A detailed description of our search strategy can be found in supplementary appendix (S1 Fig). Identified records from the electronic searches were downloaded and imported into Covidence systematic review software (Veritas Health Innovation, Melbourne, Australia https://www.covidence.org/). Abstract and full text review as well as data extraction were performed in duplicate by two reviewers (S.M., J.R.).

We employed the following inclusion criteria: all pediatric and adult patients with OSA, nasal obstruction, or mouth breathing during sleep; and oral/mouth taping, or any similar devices such as oral porous patch and chinstraps. We considered randomized control trials and prospective studies only, with objective and subjective outcomes. The exclusion criteria included non-English articles, only utilizing an oral devices such as mandibular advancement devices without mouth taping, tongue retaining devices, or soft palate lifts. Studies also had to be published within the last 25 years. Initially, 120 studies were identified. After automatic removal of 34 duplicates, 86 abstracts were screened, with 24 studies undergoing full-text screening. A total of 10 studies met inclusion criteria and were included in this systematic review (Fig 1).

Our study protocol was registered with PROSPERO (International Prospective Register of Systematic Reviews) under the following identifier number CRD42024509650.

## Results

### Baseline characteristics

From the included studies, six were prospective cross sectional studies, one was a randomized control trial, one was a prospective cohort study, one was a retrospective cohort study, and the last was a prospective crossover study. Eight out of the 10 utilized either adhesive tape or a sealing device to occlude the mouth. Two of studies utilized a chin strap to hold the mouth closed. Labarca et al. utilized a mouth seal as well as a mandibular advancement device (MAD) [9]. Osman et al. utilized a mouth seal as well as a chin strap and a novel nasal spray [10]. Two studies were completed by the same first author in the same year (Jau et al. 2023).

The mean or median age of each study differed, ranging between 38–64 years of age. Sample sizes for studies ranged from 9 to 71 participants, with 233 total patients across all studies included in this systematic review. The mean or median body mass index (BMI) ranged from 24 to 35. Mean or median baseline apnea-hypopnea indexes (AHI) were 13–47. Inclusion and exclusion criteria varied but four studies (Lee et al., Huang et al, Labarca et al, Osman et al), excluded patients with any form of nasal obstruction [2,4,9,10]. Only two studies (Lee et al, Huang et al) included patients with AHI less than 15. Other baseline characteristics are summarized in Table 1.

### Primary outcomes

Primary outcomes are summarized in Table 2. Six studies assessed AHI before and after implementation of their form of oral occlusion. Only two of these studies (Lee et al. and Huang et al.) reported a significant decrease in AHI post-occlusion [2,4]. Lee et al. reported a significant reduction in median AHI from 8.3 to 4.7 per hour after tape, and Huang et al. reported a statistically significant reduction in median AHI from 12 to 7.8 per hour after oral patch [2,4]. Three (Bhat et al., Labarca et al., and Osman et al.) did not detect a significant change in AHI [9–11]. Labarca et al. also compared the AHI in patients

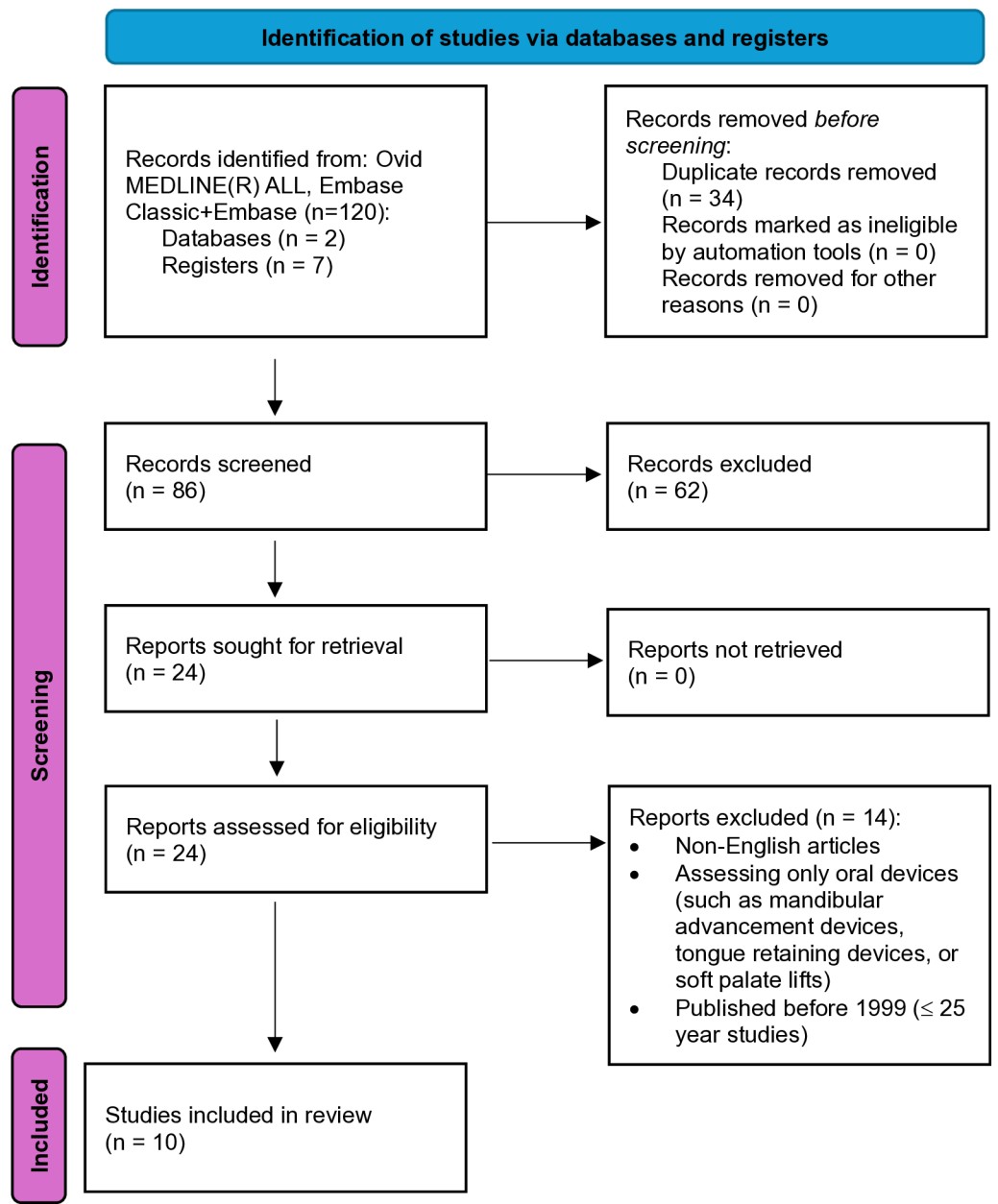

**Fig 1. Preferred Reporting Items for Systematic Reviews and Meta-Analyses (PRIMSA) flow diagram for systematic reviews.**

with MAD plus mouth taping to those utilizing just MAD alone and did report a statistically significant decrease in the median AHI with MAD alone compared to combined treatment from 10.5 to 5.6 per hour [9]. Osman et al. showed no significant difference between patient on a placebo spray compared to mouth taping. There was also no significant difference between the novel nasal spray plus mouth taping to spray alone [10]. The last study looking at AHI was one of the two studies by Jau et al. This group assessed and validated a "mouth puffing" device while patients' mouths were taped. Jau et al. define mouth puffing as a derivation of mouth breathing while patients' mouths are occluded with tape [1,12]. As such, they utilized accelerometers on the sides of both cheeks to detect the mouth puffing phenomenon while patients' mouths were

**Table 1. Baseline characteristics of selected studies.**

| Study | Intervention type | Study Type | Mean/ Median Age | Sample Size | Inclusion Criteria | Exclusion Criteria | mean/ median BMI | Additional Baseline Characteristics |
|---|---|---|---|---|---|---|---|---|
| Lee at al. 2022 | Mouth seal | Retrospective Cohort Study | median: 38 | 20 | 20-60 years of age | Retrognathia | median: 24.5 | |
| | | | | | BMI < 30 | Allergy to mouth tape | | |
| | | | | | AHI < 15 | Intolerance of mouth sealing | | |
| | | | | | Sleep disordered breathing symptoms | Comorbidies | | |
| | | | | | Witnessed mouth breathing during sleep | Tonsils grade 3/4 | | |
| | | | | | Dry mouth in the morning | Previous nose, palate, or tongue surgery | | |
| | | | | | | Shift workers | | |
| | | | | | | (Those with nasal obstruction received nasal sprays) | | |
| Madeiro et al. 2019 | Mouth seal | Prospective Cross Sectional Study | mean: 63 | 13 | 18-80 years of age | BMI > 40 | mean: 30.4 | Mean neck cirumference: 41 cm |
| | | | | | patients adapted to oronasal CPAP usage (>3 mo and > 4h/d usage) | Home O2 usage | | Mean ESS: 12 |
| | | | | | | | | Mean AHI: 43 |
| | | | | | | | | Mean duration of CPAP usage: 5 years |
| | | | | | | | | Mean CPAP level: 10.5 cmH2O |
| Bachour et al. 2003 | Chin strap | Prospective Cohort Study | mean: 53.7 | 15 | Observed mouth leak | | mean: 31 | Neck circumference: 42.6 cm |
| | | | | | Dry mouth in the morning | | | Mean CPAP level: 9.4 cmH2O |
| | | | | | Nasal obstruction with CPAP | | | |
| Huang et al. 2015 | Mouth seal | Prospective Cross Sectional Study | mean: 46 | 30 | Patients with snoring and mouth breathing during sleep | Palate position grade 3/4 | mean: 26.8 | Tonsil grade1/2: 19/11 |
| | | | | | AHI 5–15 | Tonsils grade 3/4 | | Uvula grade 1/2: 3/27 |
| | | | | | | Uvula grade >2 | | Palate grade 1/2: 12/18 |
| | | | | | | Allergic rhinitis | | |
| | | | | | | Chronic rhinitis | | |
| | | | | | | Septal devtiation | | |
| | | | | | | Sinonasal disease | | |
| | | | | | | Facial hair | | |
| | | | | | | BMI > 30 | | |

*(Continued)*

**Table 1.** (Continued)

| Study | Intervention type | Study Type | Mean/ Median Age | Sample Size | Inclusion Criteria | Exclusion Criteria | mean/ median BMI | Additional Baseline Characteristics |
|---|---|---|---|---|---|---|---|---|
| Teschler et al. 1999 | Mouth seal | Prospective Cross Sectional Study | mean: 64 | 9 | Discomfort due to nasal leak on home nasal bilvel ventilatory support | | mean: 24 | Mean AHI: 13 |
| | | | | | Significant mouth leak during ventilatory assistance in the lab | | | Mean FEV1: 33% predicted |
| | | | | | | | | Mean FEV1/VC: 52% |
| | | | | | | | | Mean PaCO2/ PaO2: 61/57 |
| | | | | | | | | IPAP/EPAP level: 17/6 cmH2O |
| Jau et al. 2023 | Mouth seal | Prospective Cross Sectional Study | mean: 45.01 | 71 | | Psychiatric disease | mean: 26.8 | Mean neck circumference: 39.25 cm |
| | | | | | | Neurological disorders | | Mean PSQI: 8.63 |
| | | | | | | Diabetes | | Mean ESS: 10.75 |
| | | | | | | Chronic renal diseases | | |
| | | | | | | Cancer | | |
| | | | | | | Cardiovascular diseases | | |
| | | | | | | Cigarette or alcohol addition | | |
| | | | | | | Sleep disorders | | |
| Bhat et al. 2015 | Chin strap | Prospective Cross Sectional Study | median: 48 | 26 | AHI >=5 | | median: 31 | median mallampati score: 4 |
| | | | | | | | | median neck circumference: 16.5in |
| | | | | | | | | Median percentage of total sleep time with SpO2 below 90%: 2.1 |
| | | | | | | | | CPAP/bilevel pressure level: 10/10 cmH2O |
| Labarca et al. 2022 | Mouth seal (+ mandibular advancement device) | Prospective Crossover Study | mean: 60.1 | 21 | 21-70 years of age | Current benzodiazepine, hypnotic, or opioid usage | mean: 26.81 | mean neck circumference: 17.47in |
| | | | | | BMI < 38 | Other sleep disorders - insomnia, nacrolepsy, central sleep apnea, parasomnia | | mean AHI: 24.35 |
| | | | | | Neck circumference <20in for men, < 17in for women | Failure to breathe comfortably through the nose | | |
| | | | | | AHI 10–50 | Allergy to adhesives | | |

*(Continued)*

**Table 1.** (Continued)

| Study | Intervention type | Study Type | Mean/ Median Age | Sample Size | Inclusion Criteria | Exclusion Criteria | mean/ median BMI | Additional Baseline Characteristics |
|---|---|---|---|---|---|---|---|---|
| | | | | | Patients with confirmed OSA and usage of a mandibular advancement device of any kind | | | |
| Osman et al. 2024 | mouth seal + chin strap (+ novel nasal spray) | Randomized Control Trial | mean: 59 | 10 | >=18 years of age | Impaired breathing - nasal congestion/obstruction | mean: 35 | mean AHI: 47 |
| | | | | | | | | mean neck circumference: 41 cm |
| | | | | | | | | mean ESS score: 7 |
| | | | | | | | | mean Insomnia Severity index: 9 |
| Jau et al. 2023 | mouth seal | Prospective Cross Sectional Study | mean: 43 | 18 | 23-57 years of age | Chronic diseases - psychiatric, neurological, diabetes, chronic renal diseases, cancers, and cardiovascular | mean: 26.96 | mean neck circumference: 38.55 cm |
| | | | | | OSA-associated symptoms - snoring and daytime sleepiness | Cigarette or alcohol addiction | | mean Pittsburgh sleep quality index: 7.56 |
| | | | | | | Known sleep disorders | | mean ESS score: 9.67 |

occluded with tape [1]. They found that AHI was significantly reduced in individuals who had no mouth puffing compared to both side or complete mouth puffing [1]. This study reported that AHI was highest in intermittent mouth puffing compared to both non and complete mouth puffing patients [1].

Snoring index (SI) was assessed by three studies. Lee at al., Bachour et al., and Huang et al. all reported a significant decrease in SI after mouth taping or chinstrap [2,4,13].

Oxygen desaturation index (ODI) was assessed by four studies. Two studies (Lee at al. and Jau et al.), reported a statistically significant decrease in ODI after mouth taping [4,12]. Bachour et al. did not find a significant decrease in ODI with chinstrap usage [13]. The second study by Jau et al. found a significantly lower ODI in those with their mouths taped with complete mouth puffing compared to side mouth puffing [1]. This study also reported significantly lower ODI in individuals with no mouth puffing compared to both side and complete mouth puffing. Lastly, the study reported that ODI was significantly higher in intermittent mouth puffing compared to side, complete, and non-mouth puffing patients [1].

Mean oxygen saturation was only assessed in one study (Lee at el.) with no significant difference with mouth taping [4].

Mean continuous positive airway pressure (CPAP) levels were assessed by Madeiro et al. and showed a significant reduction in pressure levels (cm $H_2O$) when comparing the oronasal and nasal CPAP plus mouth tape pressures [14].

Mouth leak was assessed by four studies. Mouth leak is defined as the air pressure that is lost from nasal CPAP because of patients opening their mouths during sleep which can cause upwards of 10–15% of pressure lost as leakage [13,15]. Bachour et al., Huang et al., and Jau et al. all reported that mouth leak was significantly reduced as a percentage of time slept or volume flow rate (L/s) after oral occlusion [2,12,13]. The second study by Jau et al. grouped mouth taped patients into mild to moderate OSA or severe OSA groups, and reported significantly more non and intermittent mouth puffing in both groups compared to normal individuals [1]. Those with severe OSA also had more non and intermittent

**Table 2. Primary outcomes for selected studies.**

| Study | AHI | Snoring Index (SI) | O2 Desat Index (ODI) | Mean Saturation | CPAP pressure | Airflow and Upper Airway Dimensions after flow route change (oronasal to nasal and vice-versa) | Airflow and Upper Airway Resistance | Potential Complications |
|---|---|---|---|---|---|---|---|---|
| Lee at al. 2022 | Median AHI: 8.3 before tape vs 4.7/hr afer tape (p=0.0002)* Supine AHI: 9.4 before tape vs 5.5/hr after tape (p=0.0001)* Non-supine AHI: 3.2 before tape vs 0.6/hr after tape (p=0.03)* | Median SI: 303.8 before tape vs 121.1 after tape (p=0.0002)* | Median ODI: 8.7 events/ hr before tape vs 5.8 events/hr after tape (p=0.0003)* | Median: 95 before tape vs 95 after tape | | | | "Mouth-taping is not recommended for moderate or severe OSA patients because it may impose dangers rather than benefits in these patients." |
| Madeiro et al. 2019 | | | | | 13cm H2O for oronasal CPAP pressure vs 12cm H2O for nasal CPAP pressure (p=0.039)* | Retropalatal area from nasal to orona-sal CPAP: 1109–624 pixels (p=0.001)* Retroglossal area from nasal to oronasal CPAP: 6677–4934 pixels (p=0.005)* Retropalatal area from oronasal to nasal CPAP: 723–1198 pixels (p=0.028)* Retroglossal area from oronasal to nasal CPAP: 4876–5617 pixels (p=0.012)* | Nasal to Oronasal Flow: Mean peak flow (L/s) significantly decreased from nasal to oronasal route (p=0.011)* After tap-ing, mean peak flow significantly increased (p=0.015)* | |
| Bachour et al. 2003 | | Mean snor-ing: 6.7 to 24% of total sleep time (TST) after chinstrap (p<0.005)* | Mean ODI at 4%: 13 to 5.7 after chinstrap (no sig change) | | | | | "As with any mouth occlusion, the possibil-ity exists of aspiration of stomach contents should the patient regurgitate during the night and be unable to expel the emesis." |

*(Continued)*

**Table 2.** (Continued)

| Study | AHI | Snoring Index (SI) | O2 Desat Index (ODI) | Mean Saturation | CPAP pressure | Airflow and Upper Airway Dimensions after flow route change (oronasal to nasal and vice-versa) | Airflow and Upper Airway Resistance | Potential Complications |
|---|---|---|---|---|---|---|---|---|
| Huang et al. 2015 | Median AHI: 12 to 7.8/hr after oral patch (p<0.01)* | Median SI: 146.7 to 40/hr after oral patch (p<0.01) | | | | | | "Although patients, on average, improve with treatment, the study does not define the safety or efficacy due to a small single-institution case series without a control group. Further large studies with multi-institution and control groups can be conducted to assess the efficacy and safety of the POP device." |
| Teschler et al. 1999 | | | | | | | | "The authors do not at this stage advocate taping the mouth for indiscriminate long-term home use, because of the risk of asphyxia in the presence of nasal obstruction, machine or power failure, or regurgitation." |
| Jau et al. 2023 | | | Mean ODI: 16.3 to 30.5 events/hr after tape (p = 0.037)* | | | | | |
| Bhat et al. 2015 | Median AHI: no significant difference after chinstrap (although patients on optimal CPAP settings compared to diagnostic PSG settings had significant improvement of AHI) | | | | | | | |

*(Continued)*

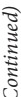

Table 2. (Continued)

| Study | AHI | Snoring Index (SI) | O2 Desat Index (ODI) | Mean Saturation | CPAP pressure | Airflow and Upper Airway Dimensions after flow route change (oronasal to nasal and vice-versa) | Airflow and Upper Airway Resistance | Potential Complications |
|---|---|---|---|---|---|---|---|---|
| Labarca et al. 2022 | Median AHI in MAD+tape compared to MAD alone: 5.6 vs 10.5/hr (p=0.02)* No significant difference comparing tape to baseline alone | | | | | | | |
| Osman et al. 2024 | No significant difference comparing placebo vs spray+tape No significant difference comparing spray+tape vs spray alone | | | | | | | |
| Jau et al. 2023 | Mean AHI: AHI/min was significant reduced during non-mouth puffing (NMP) compared to side mouth puffing (SMP) or complete mouth puffing (CMP) (p<0.001 for both)* Intermittent mouth puffing (IMP) mean AHI: AHI/min was significant higher in IMP compared to both NMP and CMP (p<0.001 for both)* | | Mean ODI: ODI/min was significantly lower in CMP compared to SMP (p<0.001)* Significantly lower in NMP compared to SMP and CMP (p<0.001 for both)* Significantly higher in IMP compared to SMP, CMP, and NMP (p<0.001 for all)* | | | | | |

| Study | Mouth leak | Arousal Index | Cephalometry | Transcutaneous CO2 Tension (Ptc,CO2) | Sleep state/REM sleep/Total sleep time | Correlation Studies | Percentage of SpO2 under 90% (T90) | |
|---|---|---|---|---|---|---|---|---|
| Lee at al. 2022 | | | | | | | | |
| Madeiro et al. 2019 | | | | | | | | |

(Continued)

Table 2. (Continued)

| Study | AHI | Snoring Index (SI) | O2 Desat Index (ODI) | Mean Saturation | CPAP pressure | Airflow and Upper Airway Dimensions after flow route change (oronasal to nasal and vice-versa) | Airflow and Upper Airway Resistance | Potential Complications |
|---|---|---|---|---|---|---|---|---|
| Bachour et al. 2003 | Mean mouth leak: 42.9% to 23.8% total sleep time (TST) after chinstrap (p<0.05)* | Mean arousal index: 33.4 to 23.6/sleep hour after chin strap (p<0.05)* | Ant lower facial height: 79.3 to 70.4mm after chinstrap(p<0.05)* Nasion and menton distance: 133.8 to 125mm after chinstrap (p<0.05)* Ant cranial baseline and mandibular line angle: 36 to 32.8 degrees after chinstrap (p<0.05)* Base of tongue and post pharyngeal wall distance: 7.8 to 9.7mm after chinstrap (p<0.05)* Tip of uvula and post pharyngeal wall distance: 2.3 to 6.4mm after chinstrap (p<0.05)* | | | | | |
| Huang et al. 2015 | | | Retropalatal space: 7.4 to 8.6mm after oral patch (p<0.05)* Retrolingual space: 6.8 to 10.2mm after oral patch (p<0.05)* | | | | | |
| Teschler et al. 1999 | Median: 0.35 to 0.06L/s after tape (p=0.05)* | No sig change | | No sig change | Rem sleep %: 12.9 to 21.1% after tape (p=0.0016)* | | | |
| Jau et al. 2023 | Mean mouth puffing: Mean intermittent mouth puffing changed from 19.06 to 26.47% (p=0.037)* | | | | | Intermittent mouth puffing was positively associated with uvula length: 0.301 (p<0.05)* ODI was negatively associated with the min width of the airway and nasal width: -0.357 and -0.381, respectively (p<0.05)* T90 was negatively associated with the min width of the airway and nasal width: -0.474 and -0.316, respectively (p<0.001)* | | |

(Continued)

Table 2. (Continued)

| Study | AHI | Snoring Index (SI) | O2 Desat Index (ODI) | Mean Saturation | CPAP pressure | Airflow and Upper Airway Dimensions after flow route change (oronasal to nasal and vice-versa) | Airflow and Upper Airway Resistance | Potential Complications |
|---|---|---|---|---|---|---|---|---|
| Bhat et al. 2015 | | | | | REM sleep % comparing diagnostic PSG to chinstrap: 20.2 to 8.7% (p = 0.0025)* REM sleep % comparing optimal CPAP to chinstrap: 25.5 to 8.7% (p = 0.0025)* Total sleep time (TST) comparing diagnostic PSG to chinstrap: 270 to 136.3mins (p < 0.001) TST comparing optimal CPAP to chinstrap: no signifcant change | | No significant difference in SpO2 nadir after chinstrap | |
| Labarca et al. 2022 | | | | | | | | |
| Osman et al. 2024 | | Mean Arousal Threshold comparing spray + tape vs placebo: 142 vs 130/hr (p < 0.05)* | | | | | | |
| Jau et al. 2023 | Mild-Mod OSA: Significantly more NMP and IMP % compared to normal individuals (p < 0.001 for both)* Severe OSA: Significantly more NMP and IMP% compared to mild-mod OSA and normal individuals (p < 0.001 for all)* | | | | | | | |

mouth puffing compared to mild to moderate OSA patients [1]. This study by Jau et al., also reported significant correlations – a positive correlation between intermittent mouth puffing and uvula length, a negative correlation between ODI and the minimal width of the airway and nasal width, and a negative correlation between the percentage of oxygen saturation (SpO2) under 90% (T90) with the minimal width of the airway and nasal width [1].

Arousal threshold, as defined by the maximum estimated ventilatory drive just before coirtical arousal during NREM (non-rapid eye movement sleep) respiratory events that resulted in arousal, was assessed by Osman et al. Patients in the novel upper airway muscle dilator spray plus tape group compared to placebo had a significantly increased arousal index with reduced events per hour [10]. The study does comment that the spray is an effective upper airway dilator, but the effects may be impaired when the mouth is taped, due to an increased sleep drive secondary to OSA [10].

Transcutaneous carbon dioxide ($CO_2$) tension (Ptc,$CO_2$), was assessed by Teschler et al., and there was no significant difference with mouth taping [16].

Two studies assessed rapid eye movement (REM) sleep states. Teschler et al. showed that REM sleep percentage significantly increased after mouth tape [16], whereas Bhat et al. showed that REM sleep percentage significantly reduced with chinstrap when compared to diagnostic polysomnography (PSG) study [11]. Although this study only performed chin strapping for the first two hours, the reduction in REM sleep with chinstrap compared to PSG was analyzed as a percentage of the total sleep time [11]. Bhat et al. also demonstrated that REM sleep percentage was significantly lower with patients using chinstraps compared to those on optimal CPAP [11]. Total sleep time was also significantly lower with chinstrap compared to patients during their diagnostic PSG study, although this was an expected result as chin strapping was only used for the first two hours [11]. Total sleep time was unchanged when comparing chinstrap to those on optimal CPAP [11]. Bhat et al. also showed that there was no significant difference in SpO2 nadir after chinstrap usage [11].

Utilizing the Newcastle-Ottawa Scale for assessing the quality and risk of bias of these 10 studies showed that all studies on mouth taping were of poor quality for varying reasons (Table 3) [17].

Table 4 summarizes the secondary/qualitative outcomes assessed by the selected studies. Huang et al. reported that median Epworth sleepiness scale (ESS) and visual analog scale of snoring (VAS) both significantly reduced after mouth sealing [2]. Labarca et al. showed no significant difference in ESS with oral tape and MAD [9].

## Discussion

Overall, the evidence surrounding the effectiveness of mouth occlusion or chin strapping is uncertain, and conclusions vary in the included studies. PSGs are considered the gold standard for the diagnosis of OSA, and although criticized for its limitations, AHI is a well-established and studied metric of OSA severity [18,19]. Only six of the ten included studies assessed AHI as a primary outcome, and only five of those compared AHI directly between interventions. Only three of these studies showed a statistically significant reduction in AHI with intervention, while two reported no significant differences. One of the included studies that reported improved AHI (Labarca et al.) only found this AHI reduction when comparing MAD plus mouth taping to MAD alone [9]. When looking into the patient subgroup with mild OSA (defined by AHI < 15), utilization of MAD or MAD plus mouth taping did show significant reduction of AHI. When this study examined baseline AHI compared to mouth tape alone, there was no significant difference in AHI [9]. Other primary outcomes utilized by authors, such as snoring index, ODI, CPAP pressures, and mouth leak, showed significant differences favoring oral occluding strategies, however, many of these primary outcomes were shown in two or fewer studies. The clinical significance of these differences is also unclear, as none represents a gold standard metric of OSA severity.

Lee et al., Huang et al., and Labarca et al. were three of the six studies showing a level of reduction in AHI with oral occluding devices [2,4,9]. However, when looking closer into these studies, Lee et al. and Huang et al. only included individuals with AHIs of less than 15 [2,4]. The classification varies between studies, however, classically, an AHI of 5–15 is considered mild OSA [18]. The AHI improvement in the study by Lee et al. was 8.3 to 4.7 (mild to borderline mild), and 12 to 7.8 in the study by Huang et al (mild to persistently mild). With the criticism of AHI as a marker for disease severity

**Table 3. Risk of bias assessment (Newcastle-Ottawa Quality assessment Scale Criteria).**

| Study | Selection | | | | Comparability | Outcome | | | Score |
|---|---|---|---|---|---|---|---|---|---|
| | Representativeness of Exposed Cohort | Selection of Non-Exposed Cohort from Same Source as Exposed cohort | Ascertainment of Exposure | Outcome of Interest was Not Present at Start of Study | Comparability of Cohorts | Assessment of Outcome | Follow-up Long Enough for Outcome to Occur (Median Duration of Follow-up≥6 months) | Adequacy of Follow-Up | Quality Score |
| Lee et al. 2022 | Participants were a selected group of patients from 2020–2021 at Chang Gung Memorial Hospital, Taiwan with inclusion and exclusion criteria | Yes ★ | Medical records/visits ★ | Yes ★ | What cofounders were adjusted for was not clearly stated | Record linkage – home sleep studies ★ | No follow-up | No | Poor |
| Madeiro et al. 2019 | Participants were of a somewhat representative group from 18–80 years of age, well adapted to oronasal CPAP, and those of BMI >40 excluded ★ | Yes ★ | Medical visit/formal sleep study ★ | Yes ★ | What cofounders were adjusted for was not clearly stated | Record linkage – formal sleep studies ★ | No follow-up | No | Poor |
| Bachour et al. 2003 | Participants were somewhat representative including patients with observed mouth leak, mouth dryness, and nasal obstruction with CPAP ★ | Yes ★ | Medical visit/formal sleep study ★ | Yes ★ | What cofounders were adjusted for was not clearly stated | Record linkage – formal sleep studies ★ | Yes ★ | Only 6/15 continued with follow-up as suitable candidates, and of those 6, only 4 completed the 6 month follow up | Poor |
| Huang et al. 2015 | Participants were of a selected group with specific inclusion and exclusion criteria | Yes ★ | Medical visit/formal sleep study ★ | Yes ★ | What cofounders were adjusted for was not clearly stated | Record linkage – formal sleep studies ★ | No follow-up | No | Poor |
| Teschler et al. 1999 | Participants were somewhat representative with specific inclusion criteria ★ | Yes ★ | Medical visit/formal sleep study ★ | Yes ★ | What cofounders were adjusted for was not clearly stated | Record linkage – formal sleep studies ★ | No follow-up | No | Poor |
| Jau et al. 2023 | Participants were truly representative ★ | Yes ★ | Medical visit/home sleep study ★ | Yes ★ | The study did control for BMI and Angle's Classification with ANOVA testing ★ | Record linkage – home sleep studies ★ | No follow-up | No | Poor |
| Bhat et al. 2015 | Participants were truly representative ★ | Yes ★ | Medical visit/formal sleep study ★ | Yes ★ | All groups were required to be normally distributed with ANOVA and Student Newman-Keuls testing ★ | Record linkage – formal sleep studies ★ | No follow-up | No | Poor |

*(Continued)*

**Table 3.** (Continued)

| | Selection | | | Comparability | Outcome | | Score |
|---|---|---|---|---|---|---|---|
| Labarca et al. 2022 | Participants were of a selected group with specific inclusion and exclusion criteria | Yes ★ | Medical visit/formal sleep study ★ | The study controlled for baseline characteristic differences using nonparametric statistical tests including Mann-Whitney U and Fisher exact tests ★ | Record linkage – formal sleep studies ★ | No follow-up | No | Poor |
| Osman et al. 2024 | Participants were truly representative ★ | Yes ★ | Blinded formal sleep study assessment ★ | The study had equal male and female participants ★ | Blinded assessment ★ | No follow-up | No | Poor |
| Jau et al. 2023 | Participants were of a selected group with specific inclusion and exclusion criteria | Yes ★ | Medical visit/formal sleep study ★ | What cofounders were adjusted for was not clearly stated | Record linkage – formal sleep studies ★ | No follow-up | No | Poor |

**Table 4. Qualitative outcomes for selected studies.**

| Study | ESS (epworth sleepiness scale) | VAS (visual analog scale of snoring) | Dry Mouth |
|---|---|---|---|
| Huang et al. 2015 | Median ESS: 8.1 to 5.2 after oral patch (p < 0.05) | Median VAS: 7.5 to 2.4 after oral patch (p < 0.05) | |
| Labarca et al. 2022 | No significant difference | | |

in OSA, it is unclear whether these reported significant reductions in AHI are meaningful clinically [20,21]. Lee et al. and Huang et al., were able to report a significant reduction of SI alongside this AHI reduction but no other clinical factors or symptoms can be commented on. Labarca et al. included a larger range of individuals with AHIs from 10–50 [9]. However, their study only reported significant reductions in AHI for those individuals that were utilizing a MAD plus mouth tape compared to those with MAD alone, while comparing oral tape to baseline AHI showed no significant difference [9]. Although, when looking at subgroup analysis in the study by Labarca et al., those with mild OSA (AHI > 15), did have significant improvement in AHI with MAD or MAD plus mouth taping [9]. These three studies also excluded any individuals with any form of nasal obstruction including allergic rhinitis, chronic rhinitis, septal deviation, sinonasal disease as well as tonsils of grade three or above [2,4,9]. Therefore, it would be fair to assume that of the patients selected for these studies, any form of oral occlusion, would allow them to continue to breathe comfortably through their nose when asleep. However, the danger arises with the trend of mouth taping in those individuals who sleep with their mouths open when baseline nasal obstruction or nasal pathology is an underlying reason. Lee at al., also mention in their discussion that mouth taping is not recommended in patients with moderate to severe OSA as it may impose dangers rather than benefits in these groups of patients [4]. Huang et al. 2015, also discuss that the safety or efficacy of oral occlusion/taping cannot be elucidated from their study given the single small institution case series without a control group [2].

With respect to risks of mouth taping, there was explicit discussion in four out of ten of the studies indicating that oral occlusion either through taping, sealing, or chin strapping could pose a serious risk of asphyxiation in the presence of nasal obstruction or regurgitation (Table 2) [2,4,13,16]. Therefore, the social media phenomena of mouth taping as a means to stop mouth breathing would seem to be guided by poor evidence and can even lead to risk of detrimental effects in individuals with serious nasal obstruction as a cause of oral breathing.

Other primary outcomes measured by three different studies were anatomic measurements by Madeiro et al., Bachour et al., and Huang et al. All three studies demonstrated that with ceasing oral breathing, oropharynx spacing increased significantly [2,13,14]. These findings align with the study by Hsu et al. 2021, who assessed patients undergoing drug-induced sleep endoscopy (DISE) and saw that oral breathing was associated with a higher degree and prevalence of lateral pharyngeal and tongue base collapse [22]. Hsu et al. 2021, theorize that the mechanism behind this is three part – the first arm is the association of mouth breathing and higher upper airway resistance compared to nasal breathing, leading to increased obstructive apneas and hypopneas [23]. The second arm is that mouth breathing decreases airway mucosa moisture and increases oropharyngeal wall surface tension causing difficulty reopening the upper airway [24]. The third arm is that nasal breathing activates nasal receptors that maintain spontaneous ventilation and oropharyngeal muscle tone which is also important for genioglossal activity [22,25]. However, both Bachour et al., and Huang et al. used cephalometric radiography in awake patients lying supine for their anatomical measurements [2,13]. Madiero et al. was the only study that assessed the oropharynx anatomy while patients were asleep using a pediatric bronchoscope [14].

Yang et al., similarly, studied patients with OSA undergoing DISE to assess total inspiratory flow in open and closed mouth settings [26]. Their group looked at 54 patients with a median AHI of 26.9 and median BMI of 28.9. Their study found that for their 32 patients with moderate levels of mouth breathing (oral airflow equating to 0.05–2.20 L/min), mouth closure increased inspiratory airflow by 2.0 L/min. For their 10 patients with near-zero mouth breathing (<0.05 L/min), mouth closure had no significant change to inspiratory airflow (0.9 L/min). For their 12 patients with high levels of mouth breathing (>2.2 L/min), mouth closure was actually detrimental and decreased airflow by -1.86 L/min [26]. When assessing

the anatomy of patients, upstream collapse was associated with greater mouth breathing and a negative response to mouth closure, especially with anteroposterior velum/soft palate collapse and concentric soft palate collapse [26]. These findings again seem to corroborate with the findings with Madeiro et al., Bachour et al., that oropharynx spacing and airflow volume increases when the mouth is closed during sleep [13,14,26]. Furthermore, Azarbarzin et al., were able to demonstrate in individuals with OSA, palatal prolapse into the velo/nasopharynx and causing expiratory flow limitation and a compensatory shunting of airflow through the mouth [27]. Taken all together, this could provide additional explanation as to why the mouth puffing phenomenon was found in the two studies by Jau et al., and that those with palatal collapse/prolapse would not benefit from mouth taping [1,12,27]. Therefore, for specific patient populations there is upstream or soft palate obstruction, mouth taping would appear to be an ineffective treatment. Additionally, Yang et al., have shown that for certain patients, namely those with high baseline mouth breathing during sleep and palatal obstruction, there are detrimental effects with forced mouth closure and that forced mouth closure during sleep is not universally beneficial [26].

The major limitations of our study include the heterogeneity of our studies, the poor quality of data based on the Newcastle-Ottawa Scale, and the limited number of studies. While we tried to make meaningful interpretations and conclusions with the small number of studies, it remains a fact that the results were very heterogenous and unfortunately statistical analysis could not be performed. Therefore, there needs to be more studies and higher quality studies to provide conclusive evidence of the safety and efficacy of this practice.

Overall, the summative data from the identified studies of this systematic review do not lend strong support to the idea of utilizing mouth taping or other occlusive devices for the improvement of OSA. All studies were of poor quality for different reasons as per the Newcastle-Ottawa assessment scale. Therefore, taking all the data together, there does seem to be a very specific use-case scenario in patient population where OSA is mild that mouth taping or occlusion may improve AHI. However, in other patient populations with nasal obstruction as a cause of mouth breathing or more severe forms of OSA, there is little evidence to support any clinical benefit for this practice. Moreover, the data identifies potential risk associated with the practice of oral occlusion for mouth breathing, SBD, or OSA.

## Conclusion

This systematic review reviewed 10 studies looking into different forms of oral occlusion in the setting of OSA or mouth breathing. Some studies report very minor improvement in certain outcomes such as AHI, ODI, and snoring index. However, the evidence for mouth taping as a treatment modality for mouth breathing, OSA, or SDB is minimal in most patient population groups outside of mild OSA, and not clinically signficiant. Moreover, there are potential serious detrimental health outcomes to those with nasal obstruction who seek oral taping as means to ameliorate their mouth breathing, OSA, or SDB during sleep. The existing data does not support mouth taping or oral occlusion as a sound clinical intervention for the general population with sleep disordered breathing.

## Supporting information

**S1 Fig. Search strategy utilized for the systematic review.**
(DOCX)

## Author contributions

**Conceptualization:** Jess Rhee, M. Elise Graham, Brian Rotenberg.

**Data curation:** Jess Rhee, Alla Iansavitchene, Sonya Mannala.

**Formal analysis:** Jess Rhee.

**Investigation:** Jess Rhee.

**Methodology:** Jess Rhee, Alla Iansavitchene, M. Elise Graham, Brian Rotenberg.

**Project administration:** Brian Rotenberg.

**Resources:** Jess Rhee.

**Software:** Alla Iansavitchene.

**Supervision:** M. Elise Graham, Brian Rotenberg.

**Validation:** Jess Rhee.

**Visualization:** Jess Rhee.

**Writing – original draft:** Jess Rhee, Sonya Mannala.

**Writing – review & editing:** Jess Rhee, Alla Iansavitchene, M. Elise Graham, Brian Rotenberg.

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
