## [Decision Letter · Decision Letter 0]

26 Dec 2024

PONE-D-24-45758Breaking Social Media Fads and Uncovering the Safety and Efficacy of Mouth Taping in Patients with Mouth Breathing, Sleep Disordered Breathing, or Obstructive Sleep Apnea: A Systematic ReviewPLOS ONE

Dear Dr.  Rotenberg,

Thank you for submitting your manuscript to PLOS ONE. After careful consideration, we feel that it has merit but does not fully meet PLOS ONE’s publication criteria as it currently stands. Therefore, we invite you to submit a revised version of the manuscript that addresses the points raised during the review process. We think that particular attention should be paid to review 3  the comments, and authors are encouraged to conduct meta-analyses, if possible.

We look forward to receiving your revised manuscript.

Kind regards,

Yongzhong Guo, Ph.D

Academic Editor

PLOS ONE

4. As required by our policy on Data Availability, please ensure your manuscript or supplementary information includes the following:

Reviewers' comments:

Reviewer's Responses to Questions

**Comments to the Author**

1. Is the manuscript technically sound, and do the data support the conclusions?

Reviewer #1: Yes

Reviewer #2: Yes

Reviewer #3: Partly

2. Has the statistical analysis been performed appropriately and rigorously? 

Reviewer #1: Yes

Reviewer #2: N/A

Reviewer #3: N/A

3. Have the authors made all data underlying the findings in their manuscript fully available?

Reviewer #1: Yes

Reviewer #2: No

Reviewer #3: Yes

4. Is the manuscript presented in an intelligible fashion and written in standard English?

Reviewer #1: Yes

Reviewer #2: Yes

Reviewer #3: Yes

5. Review Comments to the Author

Reviewer #1: The article “Breaking Social Media Fads and Uncovering the Safety and Efficacy of Mouth Taping in Patients with Mouth Breathing, Sleep Disordered Breathing, or Obstructive Sleep Apnea: A Systematic Review” explores the rising social media trend of nighttime mouth taping, presenting an evidence-based assessment of its safety and efficacy. The study adopts a robust methodological framework adhering to PRISMA guidelines and includes a systematic review of ten studies encompassing 213 patients. Despite the rigorous approach, the findings are notably heterogeneous, with limited clinical significance and potential risks outweighing benefits. While two studies showed statistically significant improvements in apnea-hypopnea index (AHI) or oxygen desaturation metrics, others either failed to demonstrate meaningful changes or highlighted risks such as asphyxiation in the presence of nasal obstructions. Notably, the article underscores the poor quality of existing research, as evaluated through the Newcastle-Ottawa Scale, which flags significant limitations in study designs and potential biases. Moreover, the exclusion of participants with nasal obstructions or severe OSA in most studies raises questions about the generalizability of the results, particularly when the practice is being marketed to a broader audience on social media. This comprehensive review highlights the lack of substantial evidence supporting the clinical utility of mouth taping for obstructive sleep apnea or mouth breathing and calls for cautious interpretation of results, advocating further high-quality research to evaluate its safety and efficacy thoroughly. The authors effectively dismantle the unfounded claims of social media trends, emphasizing the potential dangers of unregulated practices in sleep therapy, but the study's conclusions are hindered by the limited scope and quality of the available data, necessitating a cautious approach toward its recommendations.

Reviewer #2: This study is well-written but unfortunately I was not able to review all the tables that was mentioned in the manuscript and I think the tables are important since this is a systematic review hence will need to have a summary of all the studies that were included.

Reviewer #3: Review Comments

Summary

This systematic review evaluates the efficacy and safety of mouth taping for addressing mouth breathing, sleep-disordered breathing (SDB), and obstructive sleep apnea (OSA). It explores a growing trend popularized on social media and synthesizes the available evidence on its potential risks and benefits. While the review highlights critical concerns and provides valuable insights into current research, there are limitations in comparative insights, heterogeneity of included studies, and consideration of recent key publications.

Strengths

The manuscript has several notable strengths. First, it addresses a timely and highly relevant public health concern by analyzing the safety and efficacy of mouth taping, a practice that has gained widespread popularity despite the lack of substantial scientific validation. Second, the authors have employed a systematic methodology, adhering to PRISMA guidelines and utilizing a structured, librarian-assisted search strategy. This approach ensures comprehensive coverage of both indexed and non-indexed studies, enhancing the methodological rigor of the review. Third, the manuscript effectively highlights potential risks, such as asphyxiation and complications in individuals with nasal obstruction, which is a critical clinical consideration for practitioners who may encounter patient inquiries about this intervention. Lastly, the use of the Newcastle-Ottawa Scale to evaluate the included studies provides readers with a clear understanding of the quality and limitations of the evidence, further strengthening the manuscript's overall credibility..

Major concerns

C1. Lack of Comparative Insights:

The review misses the opportunity to compare its findings with recent studies, such as the JAMA Otolaryngology paper (Mouth Closure and Airflow in Patients With Obstructive Sleep Apnea: A Nonrandomized Clinical Trial). This study provides critical insights into how mouth closure influences airflow and sleep outcomes in controlled settings. Integrating these findings would enhance the depth and relevance of the discussion.

C2. Heterogeneity in Included Studies:

The studies included in the review vary widely in methodology, patient populations, and outcome measures. While this is noted, the conclusion that mouth taping is ineffective is not fully justified given this heterogeneity. Subgroup analyses or meta-analytical techniques could offer more nuanced conclusions. For example, Studies showing significant AHI reductions primarily included patients with mild OSA (baseline AHI < 15), such as:

o Lee et al.: Baseline AHI = 8.3, reduced to 4.7.

o Huang et al.: Baseline AHI = 12, reduced to 7.8.

These results suggest that mouth taping may be effective for mild OSA but less so for moderate-to-severe cases. The review should emphasize this distinction and explore other studies' baseline characteristics to better contextualize the findings.

C3. Recent Research Not Included:

Recent studies, such as the Harvard group's work on rostral airway obstruction and its influence on mouth taping efficacy, could significantly enhance the review (JAMA Otolaryngol Head Neck Surg. 2024;150(11):1012-1019. doi:10.1001/jamaoto.2024.3319). Including these findings would position the manuscript as a landmark paper in this field, reframing mouth taping as a targeted intervention rather than a one-size-fits-all solution.

Comments on the Results Presentation

The manuscript mentions: Jau et al. This group assessed and validated a “mouth puffing” device while patients’ mouths were taped. Jau et al. define mouth puffing as a derivation of mouth breathing while patients’ mouths are occluded with tape. As such, they utilized accelerometers on the sides of both cheeks to detect the mouth puffing phenomenon while patients’ mouths were occluded with tape1. They found that AHI was significantly reduced in individuals who had no mouth puffing compared to both side or complete mouth puffing

C4. In relation to the manuscript, could it be that "mouth puffing," as described by Jau et al., actually represents patients with palatal prolapse and expiratory flow limitation?

The manuscript mentions: Arousal index was assessed by Osman et al. Patients in the novel spray plus tape group compared to placebo had a significantly reduced arousal index with reduced events per hour.

C5. This outcome could potentially reflect the effect of mouth taping; however, given that this is a drug study, it seems more likely that the drug itself had a greater impact on the results. Wouldn’t this interpretation be more consistent with the study design?

The manuscript mentions: Bhat et al. also demonstrated that REM sleep percentage was significantly lower with patients using chinstraps compared to those on optimal CPAP12. Total sleep time was also significantly lower with chinstrap compared to patients during their diagnostic PSG study12 The cited reduction in REM sleep percentage and total sleep time (TST) with chin straps should be interpreted cautiously, as the original study only used chin straps for the first two hours of a split-night PSG protocol:

C3. The original Bhat study explicitly states: "Polysomnogram [PSG] underwent a modified split-night PSG, using only a chinstrap for the first 2 hours of sleep, followed by CPAP titration for the remainder of the night." This indicates that the diagnostic PSG was conducted as a full-night study, while the chinstrap was only used during the first 2 hours of the CPAP titration night. Given this setup, wouldn't it be inevitable for the total sleep time (TST) to be shorter during the chinstrap period and, consequently, the REM sleep percentage to differ?

Comments on the Discussion section

The manuscript mentions: Lee et al., Huang et al., and Labarca et al. were three of the six studies showing a level of reduction in AHI with oral occluding devices2,4,10. However, when looking closer into these studies, Lee et al. and Huang et al. only included individuals with AHIs of less than 152,4. The classification varies between studies, however, classically, an AHI of 5-15 is considered mild OSA18. The AHI improvement in the study by Lee et al. was 8.3 to 4.7 (mild to borderline mild), and 12 to 7.8 in the study by Huang et al (mild to persistently mild). With the criticism of AHI as a marker for disease severity in OSA, it is unclear whether these reported significant reductions in AHI are meaningful clinically20,21.

C4. As the authors demonstrated in the primary outcome section of the results, it cannot be denied that taping statistically lowers AHI in the entire cohort or specific patient subgroups. For instance, the Labarca study states: "A total of 21 participants were included (baseline AHI = 24.3 ± 9.9 events/h). The median AHI (interquartile range) in the MAD and MAD + AMT arms were 10.5 (5.4-19.6) events/h and 5.6 (2.2-11.7) events/h (P = 0.02), respectively. A total of 76% of individuals achieved an AHI of <10 events/h in the MAD + AMT arm versus 43% in the MAD arm (P < 0.01). Finally, the observed effect was similar in moderate to severe OSA (AHI ⩾ 15 events/h) in terms of absolute reduction and treatment responders, and AMT alone did not significantly reduce the AHI compared with baseline." This aligns with the authors' point that, in a randomized study, the combination of MAD + mouth tape (AMT) is more effective than MAD alone. This finding clearly demonstrates the effect of AMT in at least certain subgroups.

Additionally, “Jau et al. assessed and validated a ‘mouth puffing’ device while patients’ mouths were taped. Jau et al. define mouth puffing as a derivation of mouth breathing while patients’ mouths are occluded with tape. As such, they utilized accelerometers on the sides of both cheeks to detect the mouth puffing phenomenon while patients’ mouths were occluded with tape. They found that AHI was significantly reduced in individuals who had no mouth puffing compared to both side or complete mouth puffing.” This finding also suggests that mouth taping may worsen apnea in specific subgroups, particularly in patients with mouth puffing. Given that individuals with mouth puffing are likely to have palatal prolapse or expiratory flow limitation, wouldn’t it be reasonable to interpret this as evidence that mouth taping may be effective in subgroups without these anatomical or physiological features?

Additionally, considering that MAD and similar interventions are generally effective in patients with mild OSA, wouldn’t it be more appropriate to interpret this as evidence of effectiveness in mild patient populations, rather than stating, “it is unclear whether these reported significant reductions in AHI are meaningful clinically”? This perspective aligns better with the observed outcomes.

The manuscript mentions: The third arm is that nasal breathing activates nasal receptors that maintain spontaneous ventilation and oropharyngeal muscle tone which is also important for genioglossal activity22.

C5. his section's reference should be clarified, as reference 22 appears to be related to DISE and not to nasal receptors as stated. Could you confirm the correct reference for this claim? It is important to ensure the accuracy of cited studies to maintain the validity of the discussion.

The manuscript mentions: Other primary outcomes measured by three different studies were anatomic measurements by Madeiro et al., Bachour et al., and Huang et al.. All three studies demonstrated that with ceasing oral breathing, oropharynx spacing increased significantly2,14,15. These findings align with the study by Hsu et al. 2021, who assessed patients undergoing drug-induced sleep endoscopy and saw that oral breathing was associated with a higher degree and prevalence of lateral pharyngeal and tongue base collapse22. Hsu et al. 2021, theorize that the mechanism behind this is three part – the first arm is the association of mouth breathing and higher upper airway resistance compared to nasal breathing, leading to increased obstructive apneas and hypoapneas24. The second arm is that mouth breathing decreases airway mucosa moisture and increases oropharyngeal wall surface tension causing difficulty reopening the upper airway25. The third arm is that nasal breathing activates nasal receptors that maintain spontaneous ventilation and oropharyngeal muscle tone which is also important for genioglossal activity22. However, both Bachour et al., and Huang et al. used cephalometric radiography in awake patients lying supine for their anatomical measurements2,14. Madiero et al. was the only study that assessed the oropharynx anatomy while patients were asleep using a pediatric bronchoscope15.

C6. The conclusion in this section requires further consideration. For example, “Hsu et al. 2021, who assessed patients undergoing drug-induced sleep endoscopy and saw that oral breathing was associated with a higher degree and prevalence of lateral pharyngeal and tongue base collapse” suggests that mouth closure may have anatomical benefits. While this conclusion itself could be subject to error, it does not indicate negative effects of mouth taping. Moreover, the positive effects of mouth taping, as identified in references 2, 4, and 15, cannot be dismissed based on this evidence.

This section would benefit from a different approach. For instance, focusing on identifying subgroups of patients who benefit from mouth taping versus those who do not could lead to a more robust and clinically meaningful conclusion. Such an approach could significantly enhance the quality and relevance of the manuscript.

Conclusion

The authors have conducted a thorough and systematic analysis of the relevant data, and I agree with their observation that there is no definitive evidence supporting the efficacy of mouth taping. However, based on the authors' analysis, it seems more accurate to conclude that the effects of mouth taping are inconsistent rather than absent.

To make this manuscript more impactful, shifting the focus from merely evaluating the overall efficacy of mouth taping to other aspects might be beneficial. For example, the authors could consider summarizing the varying effects of mouth taping, proposing potential indication groups, or systematically discussing the Potential Complications outlined in Table 2. Such an approach could provide more clinical relevance and guidance, enhancing the manuscript’s contribution to the field.

Final Recommendation

Major Revisions.

This manuscript has potential but requires significant revisions to incorporate recent research, refine its conclusions, and provide actionable guidance for clinicians. By addressing these issues, the manuscript could become a key resource in discussing mouth taping and related interventions for OSA.

6. PLOS authors have the option to publish the peer review history of their article (what does this mean? ). If published, this will include your full peer review and any attached files.

**Do you want your identity to be public for this peer review?** For information about this choice, including consent withdrawal, please see our Privacy Policy .

Reviewer #1: **Yes: ** Denis Banchenko

Reviewer #2: No

Reviewer #3: No

---

## [Author Response · Author response to Decision Letter 1]

12 Feb 2025

All reviewer comments have been responded to in the corresponding document titled "Reviewer Comments Feb 3 2025".

We thank all the editors for their great insight and time with reviewing our manuscript!

---

## [Decision Letter · Decision Letter 1]

9 Mar 2025

PONE-D-24-45758R1Breaking Social Media Fads and Uncovering the Safety and Efficacy of Mouth Taping in Patients with Mouth Breathing, Sleep Disordered Breathing, or Obstructive Sleep Apnea: A Systematic ReviewPLOS ONE

Dear Dr. Rotenberg,

Thank you for submitting your manuscript to PLOS ONE. After careful consideration, we feel that it has merit but does not fully meet PLOS ONE’s publication criteria as it currently stands. Therefore, we invite you to submit a revised version of the manuscript that addresses the points raised during the review process.

Please submit your revised manuscript by Apr 23 2025 11:59PM. If you will need more time than this to complete your revisions, please reply to this message or contact the journal office at plosone@plos.org . Please include the following items when submitting your revised manuscript:

We look forward to receiving your revised manuscript.

Kind regards,

Yongzhong Guo, Ph.D

Academic Editor

PLOS ONE

Journal Requirements:

Reviewers' comments:

Reviewer's Responses to Questions

**Comments to the Author**

1. If the authors have adequately addressed your comments raised in a previous round of review and you feel that this manuscript is now acceptable for publication, you may indicate that here to bypass the “Comments to the Author” section, enter your conflict of interest statement in the “Confidential to Editor” section, and submit your "Accept" recommendation.

Reviewer #1: (No Response)

Reviewer #3: (No Response)

2. Is the manuscript technically sound, and do the data support the conclusions?

Reviewer #1: Yes

Reviewer #3: Yes

3. Has the statistical analysis been performed appropriately and rigorously? 

Reviewer #1: Yes

Reviewer #3: Yes

4. Have the authors made all data underlying the findings in their manuscript fully available?

Reviewer #1: Yes

Reviewer #3: (No Response)

5. Is the manuscript presented in an intelligible fashion and written in standard English?

Reviewer #1: Yes

Reviewer #3: Yes

6. Review Comments to the Author

Reviewer #1: (No Response)

Reviewer #3: Reviewer Comments on Revised Manuscript

Dear Authors,

I appreciate the efforts you have made in revising the manuscript and addressing the comments provided. The revisions have strengthened the manuscript, but there are several points that require further clarification and modifications. Below are my comments on the revised version:

C2. Heterogeneity in Included Studies

The heterogeneity of the included studies remains a crucial factor in interpreting the findings. I strongly recommend that this be explicitly addressed in the limitations section of the manuscript to ensure transparency regarding the variability across studies.

C4. Expiratory Flow Limitation and Palatal Prolapse

It would be beneficial to mention the potential role of expiratory flow limitation and palatal prolapse in interpreting the results. This could provide further insight into why mouth puffing occurs and why certain patient populations may not benefit from mouth taping.

Page 13: Correction of Mild OSA Definition

The statement on Page 13 currently reads:

"When looking into the patient subgroup with mild OSA (defined by AHI >15), utilization of MAD or MAD plus mouth taping did show significant reduction of AHI."

However, mild OSA is classically defined as AHI <15, not AHI >15. This should be corrected to:

"Mild OSA (defined by AHI <15)"

C4 (Page 16): Misinterpretation of Palatal Obstruction and Mouth Taping Effectiveness

The authors state:

"Therefore, for specific patient populations where AHI is mild and there is limited upstream or soft palate obstruction, there does seem to be a potential for benefit. However, interestingly, Yang et al., have shown that for certain patients, namely those with high baseline mouth breathing during sleep, there are detrimental effects with forced mouth closure and that forced mouth closure during sleep is not universally beneficial."

However, this interpretation needs correction based on the anatomical and physiological context:

• Mouth puffing is strongly associated with palatal prolapse, which leads to expiratory flow limitation. This means that mouth puffing occurs when there is palatal airway obstruction during exhalation.

• Patients with palatal prolapse typically have a longer palate, leading to palatal anteroposterior (AP) obstruction.

• Yang et al.'s study also demonstrated that patients with palatal obstruction do not benefit from mouth taping.

Thus, the findings from both studies are actually consistent, showing that patients with upper airway (palatal) obstruction do not respond well to mouth taping.

Critical Error on Page 16:

• The phrase "limited upstream or soft palate obstruction" suggests that mouth taping is beneficial for such patients, which is incorrect.

• The correct interpretation is that mouth taping is ineffective in patients with significant upstream or soft palate obstruction.

• This is a critical error that must be corrected.

C5. Validity of Referencing Non-Empirical Discussion Points

• If the statement in C5 is not based on direct research findings but rather reflects the authors' interpretation in the discussion, then citing it as a reference must be reconsidered.

• Discussion-based opinions from the original study authors should not be treated as empirical evidence.

• Please ensure that references used are derived from actual study findings rather than speculative discussion points.

Conclusion

The revised manuscript has improved, but several point should be addressed to ensure the accuracy and scientific integrity of the findings.

Key revisions needed:

Address study heterogeneity in the limitations section

Correct the definition of mild OSA (AHI <15) on Page 13

Revise the interpretation of mouth puffing and palatal prolapse on Page 16

Correct the misleading statement that "limited upstream or soft palate obstruction" improves with mouth taping

Re-evaluate the appropriateness of citing discussion-based opinions as references in C5

Once these changes are made, the manuscript will be in a much stronger position.

Best regards,

7. PLOS authors have the option to publish the peer review history of their article (what does this mean? ). If published, this will include your full peer review and any attached files.

**Do you want your identity to be public for this peer review?** For information about this choice, including consent withdrawal, please see our Privacy Policy .

Reviewer #1: **Yes: ** Denis Banchenko

Reviewer #3: **Yes: ** Hyung Chae Yang

---

## [Author Response · Author response to Decision Letter 2]

30 Mar 2025

Thank you reviewers for your time with our manuscript. We hope this is to your satisfaction and we look forward to any more additional comments. We have added the tables to the manuscript and have removed the separate table document as all tables were essential and not supplementary.

Thank you truly

---

## [Editor Report · Decision Letter 2]

14 Apr 2025

Breaking Social Media Fads and Uncovering the Safety and Efficacy of Mouth Taping in Patients with Mouth Breathing, Sleep Disordered Breathing, or Obstructive Sleep Apnea: A Systematic Review

PONE-D-24-45758R2

Dear Dr. Rotenberg,

We’re pleased to inform you that your manuscript has been judged scientifically suitable for publication and will be formally accepted for publication once it meets all outstanding technical requirements.

Kind regards,

Yongzhong Guo, Ph.D

Academic Editor

PLOS ONE
---

## [Editor Report · Acceptance letter]

PONE-D-24-45758R2

PLOS ONE

Dear Dr. Rotenberg,

I'm pleased to inform you that your manuscript has been deemed suitable for publication in PLOS ONE. Congratulations! Your manuscript is now being handed over to our production team.

Kind regards,

on behalf of

Dr. Yongzhong Guo

Academic Editor

PLOS ONE